# Vascular Biomarkers for Pulmonary Nodule Malignancy: Arteries vs. Veins

**DOI:** 10.3390/cancers16193274

**Published:** 2024-09-26

**Authors:** Tong Yu, Xiaoyan Zhao, Joseph K. Leader, Jing Wang, Xin Meng, James Herman, David Wilson, Jiantao Pu

**Affiliations:** 1Department of Bioengineering, University of Pittsburgh, Pittsburgh, PA 15213, USA; toy25@pitt.edu; 2Department of Radiology, University of Pittsburgh, Pittsburgh, PA 15213, USA; xiz318@pitt.edu (X.Z.); jklst3@pitt.edu (J.K.L.); jiw346@pitt.edu (J.W.); xim94@pitt.edu (X.M.); 3Department of Medicine, University of Pittsburgh, Pittsburgh, PA 15213, USA; hermanj3@upmc.edu (J.H.); wilsondo@upmc.edu (D.W.); 4Department of Ophthalmology, University of Pittsburgh, Pittsburgh, PA 15213, USA

**Keywords:** pulmonary nodules, macro-vasculature, malignancy, machine learning

## Abstract

**Simple Summary:**

The prevalence of indeterminate pulmonary nodule (IPN) findings in early lung cancer screening settings highlights the need for novel imaging biomarkers to assist clinicians in making informed therapeutic decisions. The surrounding vasculature of pulmonary nodules, termed “macro-vasculature”, has been identified as a potential biomarker for evaluating nodule malignancy. However, the relationship between the macro-vasculature’s arteries and veins surrounding an IPN and its malignancy remains unclear. This study aimed to investigate this association. Our findings indicate that arterial characteristics significantly outweigh venous characteristics in distinguishing malignant from benign nodules. Additionally, incorporating arterial information surrounding a nodule enhances the performance in differentiating malignant from benign nodules. Clarifying the relationship between macro-vasculature arteries and nodule malignancy may facilitate clinical follow-up procedures for screen-detected pulmonary nodules.

**Abstract:**

Objective: This study aims to investigate the association between the arteries and veins surrounding a pulmonary nodule and its malignancy. Methods: A dataset of 146 subjects from a LDCT lung cancer screening program was used in this study. AI algorithms were used to automatically segment and quantify nodules and their surrounding macro-vasculature. The macro-vasculature was differentiated into arteries and veins. Vessel branch count, volume, and tortuosity were quantified for arteries and veins at different distances from the nodule surface. Univariate and multivariate logistic regression (LR) analyses were performed, with a special emphasis on the nodules with diameters ranging from 8 to 20 mm. ROC-AUC was used to assess the performance based on the k-fold cross-validation method. Average feature importance was evaluated in several machine learning models. Results: The LR models using macro-vasculature features achieved an AUC of 0.78 (95% CI: 0.71–0.86) for all nodules and an AUC of 0.67 (95% CI: 0.54–0.80) for nodules between 8–20 mm. Models including macro-vasculature features, demographics, and CT-derived nodule features yielded an AUC of 0.91 (95% CI: 0.87–0.96) for all nodules and an AUC of 0.82 (95% CI: 0.71–0.92) for nodules between 8–20 mm. In terms of feature importance, arteries within 5.0 mm from the nodule surface were the highest-ranked among macro-vasculature features and retained their significance even with the inclusion of demographics and CT-derived nodule features. Conclusions: Arteries within 5.0 mm from the nodule surface emerged as a potential biomarker for effectively discriminating between malignant and benign nodules.

## 1. Introduction

Lung cancer claims more lives than the next three leading cancers (i.e., colon, breast, and prostate cancer) combined [1]. The efficacy of low-dose computed tomography (LDCT) scans in detecting early lung cancer has been established. LDCT screening is associated with a 20% of reduction in lung cancer-related mortality compared to chest X-rays [2,3]. However, the frequent identification of pulmonary nodules can present a significant challenge to lung cancer screening. The National Lung Screening Trial (NLST) reported that 96.4% of the nodules detected are ultimately proven to be non-cancerous or false positives [4]. This high false positive rate often leads to unnecessary follow-up procedures [5], including additional imaging and biopsy. These follow-up procedures not only exacerbate patient anxiety but also lead to unnecessary radiation exposure and additional cost. 

Significant research effort has been dedicated to identifying biomarkers that can differentiate between benign and malignant nodules. Traditional approaches have focused on clinical and radiographic features, such as nodule size, density, spiculation, calcification, and irregularity [6,7,8,9,10,11,12]. Machine learning approaches have been applied to determine the malignancy of nodules with promising results [6,8,10,13,14]. Deep learning (DL) technology has also been utilized for this purpose [15,16,17,18,19] due to its ability to implicitly learn intricate image textures and patterns. A significant challenge associated with DL approaches is the need for a large diverse dataset to effectively train an artificial intelligence (AI) model. Moreover, the inherent implicit learning nature of DL makes it challenging to elucidate the decision-making process of the AI model, hindering its ability to discern specific features associated with lung cancer. 

Our team has recently identified an association between the vasculature surrounding a pulmonary nodule (“macro-vasculature”) and the nodule malignancy [20,21]. The rationale is that tumor growth triggers the formation of new blood vessels (or angiogenesis) to increase its vascular supply. Angiogenesis occurs not only within a tumor but also extends to the surrounding region. However, the relationship between the arteries and veins surrounding a pulmonary nodule and its malignancy remains unclear. Clarifying this association may help gain valuable insights into the inherent roles played by arteries and veins in the development of lung cancer, leading to advancements in the assessment of nodule malignancy. The objective of this study is to explore the roles of the arteries and veins surrounding a pulmonary nodule in nodule malignancy assessment. Feature importance analyses were performed to identify the specific macro-vasculature features associated with nodule malignancy. Moreover, we investigated whether the inclusion of macro-vasculature features could improve the ability to differentiate between benign and malignant nodules when combined with traditional malignancy biomarkers, including demographic and nodule characteristics. A special emphasis was placed on small nodules with a diameter ranging from 8 to 20 mm, which are the most clinically challenging.

## 2. Materials and Methods

### 2.1. Study Subjects 

A dataset of 146 subjects (male: 84 (57.4%), Table 1) from an LDCT lung cancer screening program at the University of Pittsburgh Medical Center (UPMC) was used in the study. This screening program recruited volunteers between January 2002 and April 2005. The inclusion criteria were: (1) aged 50–79 years old and (2) current or former cigarette smokers with at least 12.5 pack-years at the time of enrollment. The exclusion criteria were: (1) having quit smoking >10 years earlier, (2) a reported history of lung cancer, or (3) a reported chest CT scan within one year of enrollment. The 146 subjects were selected to ensure a relatively balanced distribution of gender, age, and smoking history between the cancer and benign groups. In total, 67 subjects were diagnosed with non-small-cell lung cancer, while 79 had suspicious nodules later confirmed to be benign. The benign group (66.11 ± 4.65 years) was older than the malignant group (63.13 ± 5.97 years). The baseline CT scans of the 146 subjects were used in the study.

A total of 148 nodules were evaluated in the study, with 2 subjects having 2 nodules each. Nodule diameter sizes ranged from 2.39 to 39.50 mm, with 69 malignant and 79 benign. There were 70 nodules between 8 to 20 mm in size, with 33 malignant and 37 benign. For nodules less than 8.0 mm, 40 were benign and 2 were malignant. For nodules greater than 20.0 mm, 2 were benign and 34 were malignant. This study was approved by the University of Pittsburgh Institutional Review Board (IRB 21020128).

### 2.2. Differentiation and Quantification of Macro-Vascular Arteries and Veins 

We developed a novel AI algorithm to automatically identify lung arteries and veins on non-contrast CT scans [22] (Figure 1a,b). Deep learning and computational differential geometry technologies were combined to identify both extra- and intra-pulmonary arteries and veins. Initially, we employed a convolutional neural network (CNN) model to identify the central extrapulmonary arteries and veins. Then, a computational differential geometry method was applied to automatically detect tubular-like structures in the lungs exhibiting high densities, which we believe are intrapulmonary vessels. Thereafter, starting with the extrapulmonary arteries and veins, the intrapulmonary vessels were progressively tracked by following their skeletons, ultimately distinguishing between arteries and veins. The utilization of the computational differential geometry method allows the identification of small vessels. Our experiments on an independent test set (*n* = 15) showed no branch labeled by the radiologists was missed by the computer algorithm, and 98% of the intra-pulmonary vessel branches were correctly labeled [22]. This AI algorithm was applied to the LDCT scans in our cohort to automatically identify the lung arteries and veins [22].

The algorithm [21] for segmenting the macro-vasculature depicted on low-dose CT scans (Figure 1c,d) was based on the computational differential geometry method we developed earlier [22]. After differentiating between pulmonary arteries and veins, the arteries and veins surrounding a nodule were identified. Three metrics were computed for the arteries and veins based on their skeletonization: (1) volume, (2) branch count (number of branches), and (3) branch tortuosity (computed as the ratio of a branch’s curved length to its straight length). These three metrics capture essential aspects of vessel architecture and were evaluated at multi-scales, specifically at distances of 5, 10, and 15 mm measured from the nodule surface, excluding the interior vasculature of the nodule.

### 2.3. Nodule Features 

Our in-house automated nodule segmentation algorithm was used to segment and quantify nodule characteristics (Figure 1) [23,24]. The characteristics include volume, mean density, mean/maximum diameter, calcification ratio, solidness, cavity, surface area, and irregularity (the ratio between surface area and volume). 

### 2.4. Statistical Analysis

Univariate logistic regression was first used on unstandardized data to analyze and interpret all macro-vasculature features to classify nodules as benign or malignant. Subsequently, all numeric features underwent z-score standardization. Then multivariate logistic regression was implemented for nodule classification. To build the multivariate logistic regression (LR) classifiers, least absolute shrinkage and selection operator (LASSO) regularization was utilized to select features by shrinking and reducing certain LASSO regression coefficients to zero. This procedure was carried out with a 10-fold cross-validation approach to fine-tune the optimal regularization parameter to ensure robustness. In a subset of cases that included only nodules between 8–20 mm, 5-fold cross-validation was used. Features with LASSO coefficients equating to zero were excluded. Features with a variance inflation factor (VIF) larger than a threshold of 5.0 were removed from the model. To mitigate overfitting, forward stepwise selection was applied if the number of predictors did not align with the one-in-ten-rule [25]. A multivariate LR classifier was created with features exhibiting a *p*-value less than 0.05 considered significant. 

The final performance of the classification models was evaluated by the area under (AUC) the receiver operating characteristic (ROC) curve with 95% confidence intervals (CIs) computed using the 10-fold and 5-fold cross-validation method for the entire cohort and 8–20 mm nodule dataset, respectively. The DeLong test [26] was used to assess the statistical significance of the difference between two AUCs. All statistics were performed in Python 3.11.4 and RStudio 4.3.1. 

### 2.5. Integrative Prediction Modeling

Three classification models were developed based on artery-specific features (artery model), vein-specific features (vein model), and artery-and-vein-specific features (vessel model) of the macro-vasculature. The features were evaluated for arteries and veins within 5, 10, and 15 mm distances from the nodule surface. The model-derived features were designated “artery5_count”, “vein5_volume”, etc. 

The macro-vasculature features, demographic information, and nodule characteristics were used to construct the following models: (1) macro-vasculature model, (2) demographics model, (3) CT-derived nodule model, (4) macro-vasculature + demographics model, (5) macro-vasculature + CT-derived nodule model, and (6) composite (macro-vasculature + demographics + CT-derived nodule) model. These models were constructed for all nodules and nodules with a diameter between 8 and 20 mm.

### 2.6. Feature Importance Analysis 

Feature importance was assessed using permutation importance (PI) [27] and Shapley additive explanations (SHAP) [28]. These approaches were used to evaluate the importance of both macro-vasculature features and the features within the composite model related to nodule malignancy. 

In addition to the logistic regression (LR) classifier, two additional ML classifiers, namely the support vector machine (SVM) classifier [29] and multi-layer perceptron (MLP) classifier, were also used to generate diverse feature importance scores enabling the computation of an average importance score. The SVM classifier was developed with a linear kernel and a regularization parameter of 1.0, while the MLP classifier was trained with 100 hidden layers and a constant learning rate of 0.001 until convergence.

## 3. Results

### 3.1. Performance on the All Nodule-Size Dataset

*Age* and *BMI* were the only significant demographic features (*p* < 0.05) (Table 1). Each additional year of age was significantly associated with an 11.3% decrease in the probability of a nodule being malignant. Increased BMI was significantly associated with a decrease in nodule malignancy. Nodule malignancy was significantly associated with higher vessel count and greater vessel volume in the univariate logistic regression analysis for both the artery and vein variables at all distances from the nodule (Table 2). The tortuosity variables were not significantly associated with nodule malignancy.

The artery, vein, and vessel models using the features within a 5 mm distance from the nodules had a significantly better performance for discriminating nodule malignancy compared to the models using the features within 10 and 15 mm distances. The AUC for the Artery5, Vein5, Vessel5 models were 0.78, 0.77, and 0.78, respectively, for discriminating nodule malignancy. The models using features within 10 and 15 mm from the nodules had AUCs values ranging from 0.67 to 0.71. Artery, vein, and vessel models that integrated variables from all distances had similar performance for discriminating nodule malignancy (Table 3).

The macro-vasculature CT-derived nodule, composite, and CT-derived nodule models achieved AUC values for discriminating benign from malignant nodules that were promising, with AUCs of 0.92 (95% CI: 0.87–0.96), 0.91 (95% CI: 0.87–0.96), and 0.89 (95% CI: 0.84–0.94), respectively (Figure 2); (95% CI: 0.87–0.96) for discriminating benign and malignant nodules. The composite model included variables representing the demographic, macro-vasculature, and CT-derived nodule categories (Table 4). Nodule characteristics and artery variables were prominent in the most effective models. Compared to the macro-vasculature model, the demographics model exhibited no significant performance difference (*p* = 0.093), while the CT-derived nodule model displayed significantly higher performance (*p* = 0.004) according to the DeLong test. Additionally, with the inclusion of macro-vasculature features, the macro-vasculature + demographics model demonstrated much higher performance as compared to the independent demographics model (*p* = 0.004), the macro-vasculature + CT-derived nodule model also showed higher performance compared to the CT-derived nodule model (*p* = 0.028).

### 3.2. Performance on the 8–20 mm Nodule-Size Dataset

The artery (AUC = 0.67; 95% CI: 0.54–0.80) and vessel (AUC = 0.66; 95% CI: 0.53–0.71) models exhibited the best performance for discriminating benign from malignant nodules compared to the other models (Figure 3). Artery5_count, artery15_count, and vein10_volume were the significant variables in the top-four-performing models (Table 5).

The macro-vasculature + CT-derived nodule, composite, and CT-derived nodule models achieved similar performance for discriminating benign from malignant nodules with AUCs of 0.82 (95% CI: 0.72–0.92), 0.82 (95% CI: 0.71–0.92), and AUC of 0.80 (95% CI: 0.70–0.91), respectively (Figure 4, Table 6). In comparison to the macro-vasculature model, both the demographics model and the CT-derived nodule model showed no significant performance difference, with *p*-values of 0.656, 0.087, respectively, based on the DeLong test. Additionally, with the inclusion of macro-vasculature features, the macro-vasculature + CT-derived nodule model demonstrated higher but not significant performance than the CT-derived nodule model (*p* = 0.508).

### 3.3. Feature Importance

Artery features, especially the feature *artery5_count*, were identified as the most important indicators among all macro-vasculature features in both all-nodule-size and 8–20 mm-nodule-size models. The feature *artery5_count* retains its significance even with the inclusion of demographics and CT-derived nodule features (Table 7). The feature importance analyses of macro-vasculature included all macro-vasculature features from all bandwidths 5 mm, 10 mm, and 15 mm without applying any feature selection methods for the macro-vasculature model. Based on the average importance score, only the three most important features are listed in the tables.

## 4. Discussion

We investigated the specific associations between the arteries and veins surrounding a pulmonary nodule and malignancy. This investigation was built upon our prior study [20], which highlighted the significance of the vessels surrounding a nodule for distinguishing between benign and malignant nodules in the context of early lung cancer screening. To our knowledge, this is the first study to quantify the arteries and veins surrounding a pulmonary nodule depicted on LDCT images and to investigate their specific characteristics related to nodules’ malignancy. 

We created a case-balanced dataset from an LDCT lung cancer screening program, with 69 out of 148 nodules (46.6%) being malignant. This allowed us to explore the specific roles of arteries and veins surrounding a nodule in malignancy assessment. Our AI algorithms segmented and quantified arteries and veins at different distances from the nodule surface. We focused exclusively on the surrounding vasculature within multi-scales from the nodule surface, ensuring that the quantified metrics of surrounding arteries and veins remain independent of nodule solidity and size differences.

Univariate analysis (Table 2) demonstrated that the significant artery and vein variables were positively associated with malignancy coefficients, suggesting that increases in these features individually could potentially elevate the likelihood of nodules being malignant. This finding aligns with our previous study [20] that malignant nodules are surrounded by significantly more vessels compared to benign nodules.

The overall results of multivariate analysis (Table 3, Table 4, Table 5 and Table 6; Figure 2, Figure 3 and Figure 4) across various classification models of macro-vasculature features at different distances from the nodule surface consistently support the idea of macro-vasculature surrounding a pulmonary nodule as a biomarker for the malignancy of the nodule. Additionally, integrating macro-vasculature features with demographic and nodule characteristics improved the performance in differentiating the malignancy of nodules. 

Among the various macro-vasculature classification models and integrative classification models when combining macro-vasculature features with demographic and nodule characteristics, we identified arteries within 5.0 mm from the nodule surface (*Artery5_count*) as a crucial vascular characteristic in elucidating the connection between nodule malignancy and macro-vasculature. This particular feature played a prominent role in several well-performing models. In our previous study [20], we examined the macro-vasculatures within a fixed distance (i.e., 5 mm) from the surface of a nodule. However, we were uncertain whether this distance optimally reflects the malignancy of a nodule. Our multi-scale approach in this study provided clarity on the reasonableness of this 5 mm threshold. 

Classifiers are commonly explained through feature importance, which measures the individual contribution of each feature to a classifier’s performance [30,31]. Feature importance analyses of macro-vasculature features and the composite model provide further validation of our findings. Our variable selection approach was also validated by the feature importance analysis of macro-vasculature features without a pre-selection procedure. We conducted a comprehensive feature-importance survey using PI and SHAP importance across three different classifiers: logistic regression, SVM, and MLP. Our feature importance study consistently highlights that artery count within 5.0 mm from the nodule surface is one of the most important macro-vasculature indicators for nodule malignancy and it retains its significance even with the inclusion of demographics and CT-derived nodule features (Table 7). This finding may potentially facilitate the follow-up procedures associated with pulmonary nodules by offering clarification of the relationship between lung arteries and lung cancer development.

Arterial characteristics, included in the majority of well-performing models, exhibit a more significant impact on nodule malignancy compared to venous characteristics. The possible explanation could be that the pulmonary arteries responsible for supplying deoxygenated blood are often associated with tumor growth due to their involvement in angiogenesis—the formation of new blood vessels to support tumor growth [32,33]. As a result, pulmonary arterial characteristics may play a dominant role in models designed to distinguish between benignancy and malignancy. The positive correlation between arterial features and malignancy suggests that arterial characteristics might serve as an indicator of tumor vascularization, growth potential, and aggressiveness.

To explore the optimal distance for investigating the macro-vasculature surrounding a nodule and malignancy, we employed a multi-scale strategy to quantify macro-vasculature at distances within 5 mm, 10 mm, and 15 mm from the nodule surface. The observation of negative coefficients associated with arteries within 15 mm of the nodule surface (*Artery15_count*) in some models may seem contradictory to both our previous study [20] and the results of univariate analysis. However, this discrepancy could be attributed to a balancing act. As the vessel complexity increases and more vessels emerge farther from the nodule surface, the interplay between various factors might lead to nuanced outcomes. Examining results across all distances from the nodule surface together (5 mm, 10 mm, 15 mm, and all distances), we noticed that closer proximity to the nodule surface corresponds to better model performance. This again illustrates the correlation between spatial scales and increased vascular complexity. Notably, a smaller distance from the nodule surface may provide a more precise representation of the relationship between vascular characteristics and nodule malignancy. 

The size of a pulmonary nodule is a key factor in assessing its malignancy. Tumor mean diameter was included in most high-performance models as a primary indicator when considering CT-derived nodule features. The likelihood of nodule malignancy exhibits a positive correlation with its diameter. As the diameter increases, so does the likelihood of malignancy [34,35,36]. The higher risk of cancer is particularly notable in nodules larger than 8 mm [37,38]. We focused on smaller nodules with a size between 8 and 20 mm [39]. Despite decreased performance, our study yielded consistent findings about macro-vasculature features across both categories of nodules. 

Additionally, each additional year of age was associated with an 11.3% (1 − *e*^−1.12^ = 11.3%) decrease in the odds of nodules being malignant (Table 1). Possible explanations are (1) the small sample size, and (2) the presence of cancer to shorten an individual’s lifespan, leading to earlier mortality among lung cancer patients. As a result, older age was associated with lower odds of malignancy in our dataset. Surprisingly, an increased BMI was counterintuitively associated with decreased malignancy incidence, exhibiting an ‘obesity paradox’ noted in several previous studies on lung cancer incidence and mortality [40,41]. Possible explanations for this paradox include BMI being a poor measure of obesity, confounding factors related to smoking and reverse causation [42]. The benign group exhibited a higher average number of pack years, which could be attributed to the possibility that patients in the malignant group tend to smoke less than those in the benign group.

The primary limitation of this study is that the methodology and the results are only valid in the current study cohort (*n* = 148), raising concerns about generalizability. Validation of our findings would be strengthened by testing them on new datasets. However, it is important to emphasize that the primary focus of this study is to elucidate the roles of arteries and veins surrounding a pulmonary nodule in malignancy assessment rather than to develop a prediction model for distinguishing between malignant and benign nodules in real-world screening settings. The above-mentioned limitations should not affect our conclusion in this regard. 

## 5. Conclusions

Our study demonstrates a significant association between the macro-vasculature surrounding pulmonary nodules and their malignancy. Among the macro-vasculature features, arteries within 5.0 mm of the nodule surface consistently emerged as the most important predictors, retaining their significance even when additional features were included. This finding highlights the close relationship between the arteries surrounding a nodule and its malignancy, validating its potential as a biomarker for nodule malignancy beyond traditional nodule characteristics. Furthermore, our study demonstrates that arterial characteristics hold greater significance than vein characteristics in distinguishing between malignant and benign nodules.

## Figures and Tables

**Figure 1 cancers-16-03274-f001:**
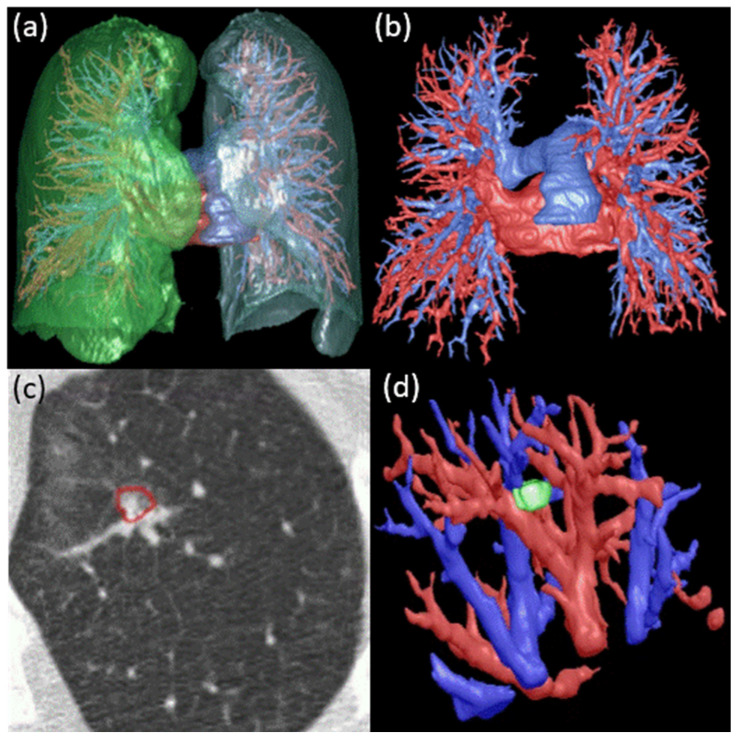
Automated segmentations of (**a**) left and right lungs, (**b**) pulmonary arteries and veins, (**c**) lung nodules, and (**d**) 3D visualization of the nodule in (**c**) and the surrounding arteries and veins.

**Figure 2 cancers-16-03274-f002:**
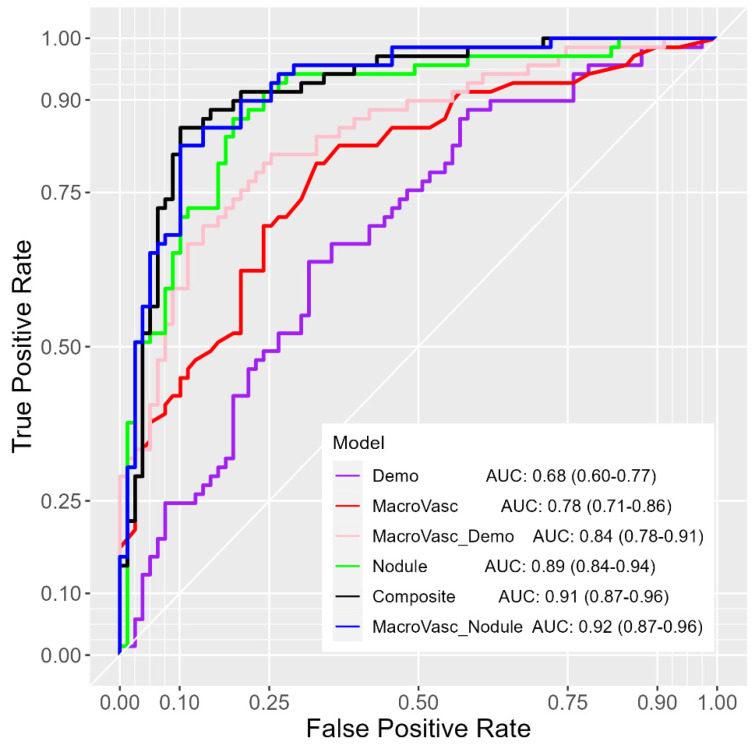
Cross-validated ROC-AUCs (95% CI) of the integrative multivariate logistic regression prediction models (*n* = 148). Demo—demographics, MacroVasc—macro-vasculature, Nodule—CT-derived nodule.

**Figure 3 cancers-16-03274-f003:**
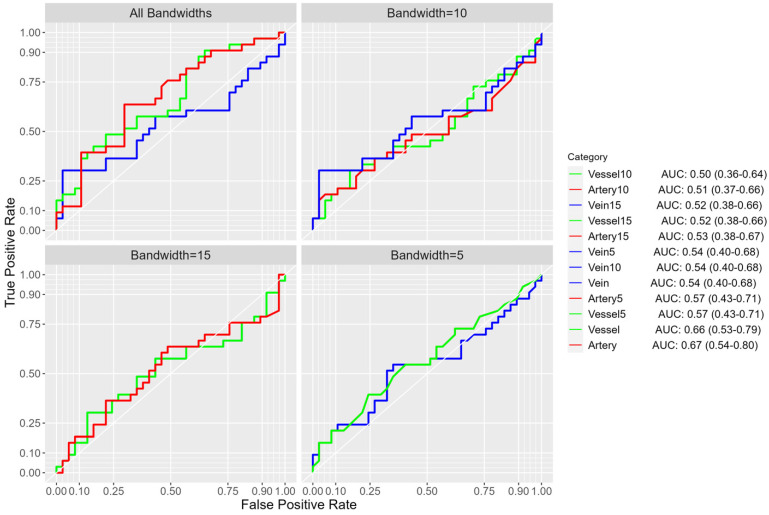
Cross-validated ROC-AUCs (95% CI) of all multivariate macro-vasculature models for 8–20 mm nodules (*n* = 70). Two ROC curves overlap if their models display the same performance.

**Figure 4 cancers-16-03274-f004:**
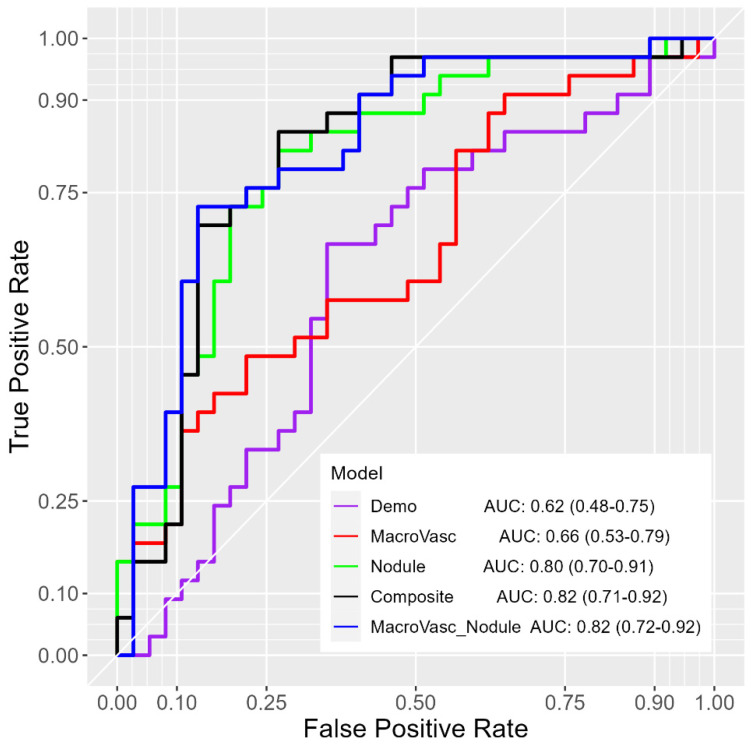
Cross-validated ROC-AUCs (95% CI) of the integrative multivariate logistic regression prediction models for 8-20 mm nodules (*n* = 70). Demo—demographics, MacroVasc—macro-vasculature, Nodule—CT-derived nodule.

**Table 1 cancers-16-03274-t001:** Subject demographics (*n* = 146).

Variable		Malignant (*n* = 67)	Benign (*n* = 79)	Coefficient (95% CI)	*p*-Value
age (year)		63.13 (5.97)	66.11 (4.65)	−0.12 (−1.20, −0.04)	0.003
gender					
	Male	36	48	−0.29 (−0.95, 0.36)	0.382
	Female	31	31		
smoking status					
	Former	43	51	−0.03 (−0.71, 0.65)	0.933
	Current	24	28		
pack (year)		45.62 (24.58)	50.33 (21.56)	−0.01 (−0.03, 0.01)	0.145
BMI		26.58 (4.14)	28.57 (4.71)	−0.10 (−0.19, −0.02)	0.011

Mean (standard deviation (SD)) and count for continuous and categorical variables respectively. Coefficient and *p*-value from the univariate logistic regression model.

**Table 2 cancers-16-03274-t002:** Univariate logistic regression model for nodule malignancy (*n* = 148).

Variable	Malignant	Benign	Coefficient (95% CI)	*p*-Value
**Artery**				
artery5_count	5.39 (4.35)	1.67 (2.18)	0.38 (0.24, 0.53)	0.000
artery5_volume	0.34 (0.61)	0.08 (0.13)	4.38 (2.16, 6.61)	<0.001
artery5_tortuosity	1.06 (0.08)	1.04 (0.16)	1.88 (−1.22, 4.99)	0.234
artery10_count	10.68 (9.67)	4.86 (4.61)	0.12 (0.06, 0.18)	<0.0001
artery10_volume	0.85 (1.21)	0.28 (0.36)	1.51 (0.71, 2.32)	<0.001
artery10_tortuosity	1.07 (0.07)	1.06 (0.15)	0.67 (−2.13, 3.47)	0.640
artery15_count	18.70 (18.26)	11.51 (10.78)	0.04 (0.01, 0.06)	0.006
artery15_volume	1.68 (2.14)	0.74 (0.89)	0.55 (0.20, 0.89)	0.002
artery15_tortuosity	1.08(0.07)	1.06 (0.14)	1.58 (−2.02, 5.17)	0.390
**Vein**				
vein5_count	5.38 (4.08)	2.17 (3.24)	0.27 (0.15, 0.39)	<0.0001
vein5_volume	0.33 (0.40)	0.09 (0.14)	4.46 (2.29, 6.64)	<0.0001
vein5_tortuosity	1.06 (0.08)	1.04 (0.14)	2.66 (−1.23, 6.55)	0.181
vein10_count	10.09 (9.10)	4.85 (5.75)	0.11 (0.05, 0.17)	<0.001
vein10_volume	0.80 (0.78)	0.27 (0.32)	1.96 (1.07, 2.85)	<0.0001
vein10_tortuosity	1.06 (0.05)	1.04 (0.14)	2.60 (−2.09, 7.28)	0.277
vein10_count	16.77 (16.01)	9.63 (8.91)	0.05 (0.02, 0.08)	0.003
vein10_volume	1.51 (1.35)	0.62 (0.62)	1.02 (0.55, 1.49)	<0.0001
vein10_tortuosity	1.06 (0.04)	1.05 (0.14)	1.54 (−2.45, 5.54)	0.449

Mean (Standard deviation (SD)) for continuous variables.

**Table 3 cancers-16-03274-t003:** Multivariate logistic regression models of macro-vasculature variables at 5, 10, and 15 mm and all distances from the nodule (*n* = 148).

**5 mm distance**			
**Variable**	**Artery5**	**Vein5**	**Vessel5**
artery5_count	**1.47**		**1.47**
vein5_count		0.60	
vein5_volume		0.70	
**AUC**	0.78 (0.71–0.86)	0.77 (0.69–0.85)	0.78 (0.71–0.86)
Accuracy	0.70 (0.62, 0.77)	0.68 (0.59, 0.75)	0.70 (0.62, 0.77)
**10 mm distance**			
**Variable**	**Artery10**	**Vein10**	**Vessel10**
artery10_count	0.54		
artery5_volume	0.67		
vein10_count		**1.24**	**1.24**
**AUC**	0.68 (0.60–0.77)	0.71 (0.62–0.80)	0.71 (0.62–0.80)
Accuracy	0.67 (0.59, 0.74)	0.68 (0.60, 0.76)	0.68 (0.60, 0.76)
**15 mm distance**			
**Variable**	**Artery15**	**Vein15**	**Vessel15**
artery15_volume	**0.91**		
vein15_volume		**1.13**	**1.13**
**AUC**	0.67 (0.58–0.76)	0.71 (0.63–0.80)	0.71 (0.63–0.80)
Accuracy	0.64 (0.56, 0.72)	0.71 (0.63, 0.78)	0.71 (0.63, 0.78)
**All distances (5, 10, 15 mm)**			
**Variable**	**Artery**	**Vein**	**Vessel**
**artery5_count**	**1.47**		**1.47**
vein5_count		0.60	
vein5_volume		0.70	
**AUC**	0.78 (0.71–0.86)	0.77 (0.69–0.85)	0.78 (0.71–0.86)
Accuracy	0.70 (0.62, 0.77)	0.68 (0.59, 0.75)	0.70 (0.62, 0.77)

Coefficients from multivariate logistic regression model. Bolded coefficients are coefficients with *p*-value < 0.05. Accuracy (95% CI) with a classification threshold of 0.5 and AUC (95% CI) are shown.

**Table 4 cancers-16-03274-t004:** Multivariate logistic regression results of multiple integrative prediction models (*n* = 148).

Model	Variables Included		Coefficient	Accuracy
Demographics	**age**		**−0.59**	0.64 (0.55, 0.71)
	**BMI**		**−0.40**	
Macro-vasculature	**artery5_count**		**1.47**	0.70 (0.62, 0.77)
CT-derived nodule	**tumor_surface_area**		**3.20**	0.80 (0.73, 0.86)
	tumor_ground_glass_ratio		0.22	
	tumor_cavitiy_ratio		−0.48	
	tumor_fat_ratio		−0.04	
	tumor_cal_volume		−0.07	
Macro-vasculature + demographics	*Demographics*	**pack/year**	**−0.72**	0.78 (0.70, 0.84)
	*Macro-vasculature*	**artery5_count**	**3.39**	
		**artery15_count**	**−1.39**	
		**vein15_count**	**−0.36**	
Macro-vasculature + CT-derived nodule	*Macro-vasculature*	**artery5_count**	**1.42**	0.84 (0.78, 0.90)
		**artery15_count**	**−1.01**	
	*CT-derived nodule*	tumor_cavitiy_ratio	−0.59	
		tumor_density	−0.35	
		tumor_irregularity	0.03	
		**tumor_mean_diameter**	**2.63**	
Composite	*Demographics*	age	−0.41	0.87 (0.81, 0.92)
		pack/year	−0.49	
		BMI	−0.27	
	*Macro-vasculature*	**artery5_count**	**0.86**	
	*CT-derived nodule*	tumor_cavitiy_ratio	−0.61	
		**tumor_mean_diameter**	**2.17**	

Bolded coefficients are coefficients with *p*-value < 0.05. Accuracy (95% CI) with a classification threshold of 0.5.

**Table 5 cancers-16-03274-t005:** Top four macro-vasculature multivariate logistic regression models in the 8–20 mm nodules subset (*n* = 70).

Variable	Artery	Vessel	Vessel5/Artery5
artery5_count	**1.60**	1.45	0.50
artery15_count	**−1.31**	**−** **1.41**	
vein10_volume		0.32	
**AUC**	0.67 (0.54, 0.80)	0.66 (0.53, 0.79)	0.57 (0.43, 0.71)
Accuracy	0.59 (0.46, 0.70)	0.60 (0.48, 0.72)	0.56 (0.43, 0.68)

Coefficients from multivariate logistic regression model. Bolded coefficients are coefficients with *p*-value < 0.05. Accuracy (95% CI) with a classification threshold of 0.5 and AUC (95% CI) are shown.

**Table 6 cancers-16-03274-t006:** Multivariate logistic regression results of multiple integrative prediction models for 8–20 mm nodules (*n* = 70).

Model	Variables Included		Coefficient	Accuracy
Demographics	age		−0.33	0.60 (0.48, 0.72)
	pack/year		−0.42	
	BMI		−0.34	
Macro-vasculature	**artery5_count**		**1.45**	0.60 (0.48, 0.72)
	**artery15_count**		**−1.41**	
	vein10_count		0.32	
CT-derived nodule	**tumor_mean_diameter**		**1.35**	0.76 (0.64, 0.85)
	tumor_AgatstonCal		−0.54	
	tumor_cavitiy_ratio		−0.67	
Macro-vasculature + demographics	*Demographics*	/	/	0.60 (0.48, 0.72)
	*Macro-vasculature*	**artery5_count**	**1.45**	
		**artery15_count**	**−1.41**	
		vein10_count	0.32	
Macro-vasculature + CT-derived nodule	*Macro-vasculature*	artery15_count	−0.81	0.77 (0.66, 0.86)
		vein10_volume	0.71	
	*CT-derived nodule*	**tumor_mean_diameter**	**1.46**	
		tumor_cavitiy_ratio	−0.57	
Composite	*Demographics*	pack/year	−0.65	0.76 (0.64, 0.85)
	*Macro-vasculature*	artery5_count	0.62	
	*CT-derived nodule*	**tumor_mean_diameter**	**1.02**	
		tumor_cavitiy_ratio	−0.79	

Bolded coefficients are coefficients with *p*-value < 0.05. Accuracy (95% CI) with a classification threshold of 0.5.

**Table 7 cancers-16-03274-t007:** Feature importance. Scores were normalized to a range of 0 to 1.

Nodule Size	Model	Features	PI Score			SHAP Score			Average Score
			LR	SVM	MLP	LR	SVM	MLP	
All-nodule-size	Macro-vasculature (3)	artery5_count	1.00	1.00	1.00	1.00	1.00	1.00	1.00
		artery15_volume	0.54	0.30	0.52	0.90	0.31	0.39	0.56
		artery15_count	0.34	0.23	0.43	0.68	0.45	0.33	0.42
	Composite	tumor_mean_diameter	1.00	1.00	1.00	1.00	1.00	1.00	1.00
		artery5_count	0.40	0.58	0.11	0.55	0.80	0.43	0.41
		tumor_cavitiy_ratio	0.22	0.22	0.05	0.18	0.25	0.00	0.17
		pack-years	0.11	0.13	0.02	0.34	0.51	0.10	0.15
		age	0.00	0.00	0.33	0.00	0.00	0.01	0.08
		BMI	0.03	0.08	0.00	0.02	0.13	0.00	0.03
8-20 mm-nodule-size	Macro-vasculature (3)	artery5_count	1.00	1.00	0.79	1.00	1.00	1.00	0.95
		vein5_volume	0.67	0.82	0.86	0.29	0.44	0.06	0.66
		artery5_volume	0.85	0.45	0.70	0.64	0.22	0.01	0.66
	Composite	tumor_mean_diameter	1.00	1.00	1.00	1.00	1.00	1.00	1.00
		pack-years	0.28	0.51	0.36	0.00	0.44	0.01	0.28
		artery5_count	0.38	0.48	0.00	0.10	0.98	0.43	0.24
		tumor_cavitiy_ratio	0.00	0.00	0.26	0.02	0.00	0.00	0.07

## Data Availability

The datasets presented in this article are not readily available because the data are part of an ongoing study.

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
