# Peer review of "Vascular Biomarkers for Pulmonary Nodule Malignancy: Arteries vs. Veins"

_cancers, 2024, doi:10.3390/cancers16193274_

Round 1

Reviewer 1 Report

Comments and Suggestions for Authors

Review of the Manuscript: “Vascular Biomarkers for Pulmonary Nodule Malignancy: Arteries vs. Veins” by Yu et al. The manuscript by Yu et al. examines the relationship between the arteries and veins surrounding a pulmonary nodule and its malignancy. The authors used 148 samples with benign and malignant lung nodules to investigate their surrounding macro-vasculature using CT scans. Features such as branch count, volume, and tortuosity of arteries and veins were quantified at varying distances from the nodule surface. The authors employed several approaches: (1) Univariate and multivariate logistic regression to identify relationships between these features and malignancy.  (2) ROC-AUC analysis to assess the performance of the models using different features. (3) Identification of features with the greatest importance for malignancy detection.

The study found that arteries within 5.0 mm of the nodule surface emerged as a potential biomarker for effectively distinguishing between malignant and benign nodules. Models incorporating macro-vasculature features, demographics, and CT-derived nodule characteristics yielded an AUC of 0.91 (95% CI: 0.87–0.96) for all nodules and an AUC of 0.82 (95% CI: 0.71–0.92) for nodules between 8-20 mm.

The study is important as it identifies promising markers for detecting malignant nodules at an early stage, potentially preventing unnecessary procedures for benign nodules. However, the manuscript requires several improvements to enhance its clarity and coherence:

1. Presentation and Flow: The manuscript should present the findings in a more cohesive and logical manner. Specifically, the feature tables and AUC curves for the models should be clearly labeled to facilitate easier understanding of the results. Additionally, several tables lack complete legends, which hinders comprehension. Including a schematic diagram summarizing the study approach would help readers better understand the methodology.

2. Data Splitting and Model Validation: It appears that the models were reported on the training dataset only. It is not clear whether the dataset of 148 samples was divided into training and testing sets. The authors should use an independent dataset to validate the models' performance. If an independent dataset is not available, they should divide 148 samples into training and testing, models should be developed using 5/10-fold cross validation, and the best models should then be tested on testing dataset.

3. Performance Metrics: The authors should report additional performance metrics such as sensitivity, specificity, accuracy, positive predictive value (PPV), and negative predictive value (NPV).

4. Authors clarify whether multivariate or univariate logistic regression was used in generating Table 3.

- Page 3, Paragraph 4: Please clarify the discrepancy in the number of samples studied, as 67 + 79 = 146, while the manuscript states that 148 nodules were evaluated. Does any sample has more than one nodule?

- Section 2.2, Page 4: Provide a more detailed explanation of the features (volume, branch count, and branch tortuosity) and the rationale for selecting them.

- Table 3: Clarify the columns and improve the legends to be more self-explanatory. For instance, indicate whether the values above the AUC row represent coefficients (such as 1.47, 0.60, 0.70).

- Table 4: The coefficient for "pack/year" is statistically significant at -0.72, which implies that increased smoking decreases the likelihood of malignancy. This result seems counterintuitive, as increased smoking is generally associated with a higher risk of lung malignancy. The authors need to justify these findings in the context of lung cancer biology with proper refernce.

- Discussion, Page 12: The statement “The possible explanation could be that the pulmonary arteries responsible for supplying deoxygenated blood are often associated with tumor growth due to their involvement in angiogenesis…” needs proper citation.

- Clarification Needed in Discussion: The meaning of the statement “…so patients with lung cancer die sooner…” is unclear. The authors should clarify the intended message, especially in the context of”. Additionally, each additional year of age was associated with a 1−𝑒−0.119 = 11.2% lower odds of nodules being malignant.”

Author Response

The manuscript by Yu et al. examines the relationship between the arteries and veins surrounding a pulmonary nodule and its malignancy. The authors used 148 samples with benign and malignant lung nodules to investigate their surrounding macro-vasculature using CT scans. Features such as branch count, volume, and tortuosity of arteries and veins were quantified at varying distances from the nodule surface. The authors employed several approaches: (1) Univariate and multivariate logistic regression to identify relationships between these features and malignancy.  (2) ROC-AUC analysis to assess the performance of the models using different features. (3) Identification of features with the greatest importance for malignancy detection.

The study found that arteries within 5.0 mm of the nodule surface emerged as a potential biomarker for effectively distinguishing between malignant and benign nodules. Models incorporating macro-vasculature features, demographics, and CT-derived nodule characteristics yielded an AUC of 0.91 (95% CI: 0.87–0.96) for all nodules and an AUC of 0.82 (95% CI: 0.71–0.92) for nodules between 8-20 mm.

The study is important as it identifies promising markers for detecting malignant nodules at an early stage, potentially preventing unnecessary procedures for benign nodules. However, the manuscript requires several improvements to enhance its clarity and coherence:

Comments 1. Presentation and Flow: The manuscript should present the findings in a more cohesive and logical manner. Specifically, the feature tables and AUC curves for the models should be clearly labeled to facilitate easier understanding of the results. Additionally, several tables lack complete legends, which hinders comprehension. Including a schematic diagram summarizing the study approach would help readers better understand the methodology.

Answer: All tables, some figures and their captions have been updated. Table 1, 2, 3, 4. 5, 6, 7. Figure 2, 3, 4.

Comments 2. Data Splitting and Model Validation: It appears that the models were reported on the training dataset only. It is not clear whether the dataset of 148 samples was divided into training and testing sets. The authors should use an independent dataset to validate the models' performance. If an independent dataset is not available, they should divide 148 samples into training and testing, models should be developed using 5/10-fold cross validation, and the best models should then be tested on testing dataset.

Response 2: Thank you for pointing this out. Unfortunately, an independent dataset is not available, which we have acknowledged as a limitation and discussed in the manuscript. Due to the relatively small sample size of our dataset, we employed k-fold cross-validation separately during both the model development and evaluation phases. We believe this approach maximizes the utility of the dataset and ensures valid results.

Comments 3. Performance Metrics: The authors should report additional performance metrics such as sensitivity, specificity, accuracy, positive predictive value (PPV), and negative predictive value (NPV).

Response 3: We apologize for this oversight. Accuracy has been reported for all models. Tables on PAGE 6, 7, and 9.

Comments 4. Authors clarify whether multivariate or univariate logistic regression was used in generating Table 3.

Response 4: We apologize for the confusion. Clarification has been updated. Table 3 on Page 6.

Comments 5. Page 3, Paragraph 4: Please clarify the discrepancy in the number of samples studied, as 67 + 79 = 146, while the manuscript states that 148 nodules were evaluated. Does any sample has more than one nodule?

Response 5: We calrify this now. Page 3, Line 109. [A total of 148 nodules were evaluated in the study, with two subjects having two nodules each.]

Comments 6. Section 2.2, Page 4: Provide a more detailed explanation of the features (volume, branch count, and branch tortuosity) and the rationale for selecting them.

Response 6: Detailed explanation and rationale have been added in section 2.2. Page 4.

Comments 7. Table 3: Clarify the columns and improve the legends to be more self-explanatory. For instance, indicate whether the values above the AUC row represent coefficients (such as 1.47, 0.60, 0.70).

Response 7: Thank you for the comment. Table 3 has been updated.

Comments 8. Table 4: The coefficient for "pack/year" is statistically significant at -0.72, which implies that increased smoking decreases the likelihood of malignancy. This result seems counterintuitive, as increased smoking is generally associated with a higher risk of lung malignancy. The authors need to justify these findings in the context of lung cancer biology with proper refernce.

Response 8: Agree. We included explanation for this finding on Page 12 Line 355. [The benign group exhibited a higher average number of pack years, which could be attributed to the possibility that patients in the malignant group tend to smoke less than those in the benign group.]

Comments 9. Discussion, Page 12: The statement “The possible explanation could be that the pulmonary arteries responsible for supplying deoxygenated blood are often associated with tumor growth due to their involvement in angiogenesis…” needs proper citation.

Response 9: We have added citations. Thank you!

Comments 10. Clarification Needed in Discussion: The meaning of the statement “…so patients with lung cancer die sooner…” is unclear. The authors should clarify the intended message, especially in the context of”. Additionally, each additional year of age was associated with a 1−?−0.119 = 11.2% lower odds of nodules being malignant.”

Response 10: Thank you for the thoughtful comment! Clarifications and revisions have been updated here. Discussion Line 346-350.

Reviewer 2 Report

Comments and Suggestions for Authors

Very interesting work. I am honored to have reviewed it. 

I only have one minor comment: 

The first 2 paragraphs of the " Discussion" chapter is rather an explained version of the results int the "Results" chapter. I would recommend to restructure the MS and include these two paragraphs (lines 20 to 274) into the "Results" section.

Comments on the Quality of English Language

The use of English language is appropriate and with very few mistakes. Easy to understand. 

line 257: "arteries and veins variables" should be "artery and vein variables"

line 349: "surrounding nodules as potential biomarkers"  should be "surrounding nodules as potential biomarker"

line 311: there is and additional space between both and our 

Author Response

Very interesting work. I am honored to have reviewed it.

I only have one minor comment:

Comments 1. The first 2 paragraphs of the " Discussion" chapter is rather an explained version of the results int the "Results" chapter. I would recommend to restructure the MS and include these two paragraphs (lines 20 to 274) into the "Results" section.

Response 1: The paragraphs in Discussion have been restructured, Page 11. Lines 274 to 291. Thank you for the comment!

The use of English language is appropriate and with very few mistakes. Easy to understand.

Comments 2. line 257: "arteries and veins variables" should be "artery and vein variables"

Response 2: We have corrected it. Line 281. Page 11.

Comments 3. line 349: "surrounding nodules as potential biomarkers"  should be "surrounding nodules as potential biomarker"

Response 3: We have revised it. Line 379. Page 13.

Comments 4. line 311: there is and additional space between both and our

Response 4: Thank you for pointing out this error. We appreciate your attention to detail!
